# Synaptopodin Regulates Denervation-Induced Plasticity at Hippocampal Mossy Fiber Synapses

**DOI:** 10.3390/cells13020114

**Published:** 2024-01-06

**Authors:** Pia Kruse, Gudrun Brandes, Hanna Hemeling, Zhong Huang, Christoph Wrede, Jan Hegermann, Andreas Vlachos, Maximilian Lenz

**Affiliations:** 1Department of Neuroanatomy, Institute of Anatomy and Cell Biology, Faculty of Medicine, University of Freiburg, 79104 Freiburg, Germany; 2Institute of Neuroanatomy and Cell Biology, Hannover Medical School, 30625 Hannover, Germany; 3Institute of Functional and Applied Anatomy, Hannover Medical School, 30625 Hannover, Germany; 4Research Core Unit Electron Microscopy, Hannover Medical School, 30625 Hannover, Germany; 5Center for Basics in Neuromodulation (NeuroModulBasics), Faculty of Medicine, University of Freiburg, 79106 Freiburg, Germany; 6Center BrainLinks-BrainTools, University of Freiburg, 79104 Freiburg, Germany

**Keywords:** lesion-induced plasticity, denervation, synaptopodin, local protein synthesis

## Abstract

Neurological diseases can lead to the denervation of brain regions caused by demyelination, traumatic injury or cell death. The molecular and structural mechanisms underlying lesion-induced reorganization of denervated brain regions, however, are a matter of ongoing investigation. In order to address this issue, we performed an entorhinal cortex lesion (ECL) in mouse organotypic entorhino-hippocampal tissue cultures of both sexes and studied denervation-induced plasticity of mossy fiber synapses, which connect dentate granule cells (dGCs) with CA3 pyramidal cells (CA3-PCs) and play important roles in learning and memory formation. Partial denervation caused a strengthening of excitatory neurotransmission in dGCs, CA3-PCs and their direct synaptic connections, as revealed by paired recordings (dGC-to-CA3-PC). These functional changes were accompanied by ultrastructural reorganization of mossy fiber synapses, which regularly contain the plasticity-regulating protein synaptopodin and the spine apparatus organelle. We demonstrate that the spine apparatus organelle and synaptopodin are related to ribosomes in close proximity to synaptic sites and reveal a synaptopodin-related transcriptome. Notably, synaptopodin-deficient tissue preparations that lack the spine apparatus organelle failed to express lesion-induced synaptic adjustments. Hence, synaptopodin and the spine apparatus organelle play a crucial role in regulating lesion-induced synaptic plasticity at hippocampal mossy fiber synapses.

## 1. Introduction

Denervation is a common feature of various neurological diseases, which are accompanied by neuronal cell death, demyelination or traumatic injury of the central nervous system [1]. As a result of the loss-of-input in otherwise healthy brain regions, complex reorganizational processes of neural circuits have been described. Specifically, excitatory synaptic strengthening upon denervation or activity deprivation has been reported as a compensatory synaptic adjustment [2,3]. However, the mechanisms that mediate lesion-induced synaptic plasticity at local dendritic sites warrant further investigation.

The hippocampal network is a crucial structure in memory acquisition. Here, the excitatory connection between dentate granule cells (dGCs) and CA3 pyramidal cells (CA3-PCs), i.e., the hippocampal mossy fiber synapse, has been demonstrated to play a crucial role in memory formation, maintenance and retrieval [4]. At this synapse, previous studies have identified distinct morphological features, such as the regular occurrence of the spine apparatus organelle, which can be found in a subset of dendritic spines in telencephalic neurons [5]. Notably, the spine apparatus organelle and its associated actin-binding molecule synaptopodin were identified as crucial regulators of synaptic plasticity [6,7,8]. Moreover, recent work suggested a potential role for synaptopodin in neuropsychiatric diseases, such as cognitive decline and autism spectrum disorders [9,10]. Although numerous studies have assessed plasticity at this synapse (e.g., the detonating transmission; [11]) and its impact on memory and behavior, no studies have addressed whether lesion-induced synaptic plasticity occurs at this synapse when the main cortical excitatory input to the hippocampus is affected. Organotypic tissue cultures of the entorhino-hippocampal complex, which contain intact cortico-hippocampal projections, provide the opportunity to study synaptic transmission at hippocampal mossy fiber synapses in a steady-state condition [12]. Based on the lamination of cortico-hippocampal inputs, the in vitro entorhinal-cortex lesion can be used to explore the effects of partial denervation on hippocampal neural circuits [13,14,15,16].

In this study we used an entorhinal cortex lesion (ECL) of organotypic entorhino-hippocampal tissue cultures to study synaptic adaptations in the dentate gyrus and CA3 network upon partial denervation. We provide evidence that partial denervation leads to ultrastructural changes and excitatory synaptic strengthening in both dGCs and CA3-PCs. Using paired whole-cell patch-clamp recordings, we demonstrate changes in synaptic transmission at hippocampal mossy fiber synapses that are accompanied by ultrastructural reorganizations of the plasticity-related spine apparatus organelle [6,8,17,18,19,20,21,22,23]. Our data suggest that synaptopodin and the spine apparatus organelle are associated with dendritic ribosomes that could be crucial for the local implementation of lesion-induced synaptic plasticity [24]. Consequently, synaptopodin-deficient tissue cultures, which lack the spine apparatus organelle, showed alterations in their ability to express lesion-induced plasticity of excitatory synapses and the region-specific transcriptome. We therefore conclude that the spine apparatus organelle is essential for regulating lesion-induced plasticity, which might be linked to the localization of ribosomes near synaptic sites.

## 2. Materials and Methods

### 2.1. Ethics Statement

Mice were maintained in a 12 h light/dark cycle with food and water available ad libitum. Every effort was made to minimize distress and pain of animals. All experimental procedures were performed according to German animal welfare legislation and approved by the appropriate animal welfare committee and the animal welfare officer of Albert-Ludwigs-University Freiburg, Faculty of Medicine (X-17/07K, X-17/09K, X-21/01B) and Hannover Medical School (2022/308).

### 2.2. Preparation of Tissue Cultures

Entorhino-hippocampal tissue cultures were prepared at postnatal days 3–5 from C57BL/6J (wildtype, Synpo^+/+^) and B6.129-Synpo^tm1Mndl/Dllr^ (Synpo^−/−^, [18,19,20]) mice of either sex as previously described [25]. Synpo^−/−^ strain was back-crossed at least 10 times to establish C57BL/6J genetic background. Cultivation medium contained 50% (*v*/*v*) MEM, 25% (*v*/*v*) basal medium eagle, 25% (*v*/*v*) heat-inactivated normal horse serum, 25 mM HEPES buffer solution, 0.15% (*w*/*v*) bicarbonate, 0.65% (*w*/*v*) glucose, 0.1 mg/mL streptomycin, 100 U/mL penicillin and 2 mM glutamax. The pH was adjusted to 7.3. All tissue cultures were allowed to mature for at least 18 days in a humidified atmosphere with 5% CO_2_ at 35 °C. Cultivation medium was replaced 3 times per week.

### 2.3. Entorhinal Cortex Lesion (ECL)

The entorhinal cortex of mature slice cultures (≥18 days in vitro) was transected with a sterile scalpel from the rhinal to the hippocampal fissure to achieve a complete lesion of the perforant path. Except for the lesion-induced partial denervation of the hippocampus, cytoarchitecture of both the hippocampus and the entorhinal cortex remained unchanged.

### 2.4. Whole-Cell Patch-Clamp Recordings

Whole-cell patch-clamp recordings were carried out at 35 °C (3–6 neurons per culture). The bath solution contained (in mM) 126 NaCl, 2.5 KCl, 26 NaHCO_3_, 1.25 NaH_2_PO_4_, 2 CaCl_2_, 2 MgCl_2_ and 10 glucose (aCSF) and was oxygenated continuously (5% CO_2_/95% O_2_). Patch pipettes contained (in mM) 126 K-gluconate, 10 HEPES, 4 KCl, 4 ATP-Mg, 0.3 GTP-Na_2_, 10 PO-Creatine, 0.3% (*w*/*v*) biocytin (pH 7.25 with KOH, 290 mOsm with sucrose), having a tip resistance of 4–6 MΩ. Neurons were visually identified using an LN-Scope (Luigs & Neumann, Ratingen, Germany) equipped with an infrared dot-contrast and a 40× water-immersion objective (NA 0.8; Olympus, Tokyo, Japan). Spontaneous excitatory postsynaptic currents (sEPSCs) of both dGCs and CA3-PCs were recorded in voltage-clamp mode at a holding potential of −70 mV. In the dentate gyrus, recordings were performed in granule cells at the outer parts of the granule cell layer in the suprapyramidal blade with maximum distance to the subgranular zone. Moreover, the resting membrane potential was estimated before our recordings to exclude immature granule cells. In CA3, recordings were performed in pyramidal cells close to the dentate gyrus in proximity to str. lucidum. Series resistance was monitored before and after each recording, and recordings were discarded if the series resistance reached ≥30 MΩ. For recording of intrinsic cellular properties in current-clamp mode, pipette capacitance of 2.0 pF was corrected and series resistance was compensated using the automated bridge balance tool of the MultiClamp commander. I-V-curves were generated by injecting 1 s square pulse currents starting at −100 pA and increasing in 10 pA steps until +500 pA current injection was reached (sweep duration: 2 s).

### 2.5. Paired Whole-Cell Patch-Clamp Recordings of Hippocampal Mossy Fiber Synapses

Action potentials were generated in the presynaptic dGC using 5 ms square current pulses (1 nA). For connectivity analysis, 5 action potentials were induced at 20 Hz (inter-sweep-interval: 10 s; at least 30 repetitions) while recording unitary excitatory postsynaptic currents (uEPSCs) from CA3-PCs. Neurons were considered to be connected if >5% of action potentials evoked time-locked inward uEPSCs [26]. Recordings were performed in the presence of (-)-bicuculline-methiodide (10 μM, Abcam, Cambridge, UK, #ab120108) to prevent inhibitory synapse recruitment. Moreover, glutamine (200 μM, Gibco, New York, NY, USA, #11539876) was added to the recording medium to avoid synaptic depletion [27].

### 2.6. Post Hoc Staining and Imaging

Tissue cultures were fixed overnight in a solution of 4% (*w*/*v*) paraformaldehyde (PFA; in PBS (0.1M, pH 7.4) with 4% (*w*/*v*) sucrose). After fixation, cultures were washed in PBS (0.1 M, pH 7.4) and consecutively incubated for 1 h with 10% (*v*/*v*) normal goat serum (NGS) in 0.5% (*v*/*v*) Triton X-100 containing PBS to reduce nonspecific staining. For post hoc visualization of patched dGCs and CA3-PCs, Streptavidin Alexa Fluor 488 (Streptavidin A488, 1:1000; Invitrogen, Waltham, MA, USA, #S32354) was added to the incubation solution. DAPI nuclear stain (1:5000 in PBS for 10 min; Thermo Scientific, Waltham, MA, USA, #62248) was used to visualize cytoarchitecture. Cultures were washed, transferred onto glass slides and mounted for visualization with DAKO anti-fading mounting medium (Agilent, Santa Clara, CA, USA, #S302380-2).

### 2.7. Serial Block-Face Scanning Electron Microscopy (SBF-SEM)

For volume electron microscopic evaluation, samples were fixed in 1.5% glutaraldehyde and 1.5% paraformaldehyde buffered in 0.15 M HEPES, pH = 7.35. Further processing was already described previously [28] based on accessible protocols (https://ncmir.ucsd.edu/sbem-protocol; accessed on 10 July 2023). In brief, for postfixation and staining reduced osmium tetroxide, thiocarbohydrazide and osmium tetroxide (rOTO), followed by uranyl acetate and lead aspartate were applied. After dehydration in an ascending acetone series, samples were embedded in epoxy resin, Araldite CY212 premix kit (Agar Scientific, Stansted Essex, UK).

Target regions were identified on semithin sections (1 μm, stained with toluidine blue). Accordingly, resin blocks were trimmed, mounted with conductive epoxy glue (Chemtronics, CircuitWorks, Kennesaw, GA, USA) on sample pin stubs (Micro to Nano, Haarlem, The Netherlands), precisely trimmed again and sputter coated with a gold layer.

Image stacks were acquired with a Zeiss Merlin VP Compact SEM (Carl Zeiss Microscopy GmbH, Oberkochen, Germany) equipped with a Gatan 3View2XP system (Gatan Inc., Pleasanton, CA, USA). Every second block-face (section thickness 50 nm) was scanned with 3.5 kV acceleration voltage and 100% focal charge compensation (10 nm pixel size, 3 μs dwell time) by using the Multi-ROI mode to record multiple areas on each surface.

### 2.8. Transmission Electron Microscopy—Mossy Fiber Synapse Visualization

Wildtype (Synpo^+/+^) and Synpo^−/−^ tissue cultures were fixed in 4% paraformaldehyde (*w*/*v*) and 2% glutaraldehyde (*w*/*v*) in 0.1 M phosphate buffer (PB) overnight and washed for 1 h in 0.1 M PB. After fixation, tissue cultures were sliced with a vibratome, and the slices were incubated with 1% osmium tetroxide for 20 min in 5% (*w*/*v*) sucrose containing 0.1 M PB. The slices were washed 5 times for 10 min in 0.1 M PB and washed in graded ethanol (10 min in 10% (*v*/*v*) and 10 min in 20% (*v*/*v*)). The slices were then incubated with uranyl acetate (1% (*w*/*v*) in 70% (*v*/*v*) ethanol) overnight and were subsequently dehydrated in graded ethanol 80% (*v*/*v*), 90% (*v*/*v*) and 98% (*v*/*v*) for 10 min. Finally, slices were incubated with 100% (*v*/*v*) ethanol twice for 15 min followed by two 15 min washes with propylene oxide. The slices were then transferred for 30 min in a 1:1 mixture of propylene oxide with durcupan and then for 1 h in durcupan. The durcupan was exchanged with fresh durcupan and the slices were transferred to 4 °C overnight. The slices were then embedded between liquid release-coated slides and coverslips. Cultures were re-embedded in blocks and ultrathin sections were collected on copper grids, at which point an additional Pb citrate contrasting step was performed (3 min). Electron microscopy was performed using a LEO 906E microscope (Zeiss) at 4646× magnification.

### 2.9. Synaptopodin Co-Immunoprecipitation (SP-coIP), RNA-Seq Analysis and Negative Contrast Electron Microscopy

Six tissue cultures per biological replicate were washed in coIP-solution and dounce homogenized on ice in a solution containing (150 μL per sample, coIP-solution, in mM) 10 HEPES, 120 NaCl, 0.05% (*v*/*v*) NP-40 or IGEPAL CA-630 (Sigma Aldrich, St. Louis, MO, USA, #I8896), cOmplete protease inhibitor (Roche, according to manufacturer’s instructions), murine RNase inhibitor (1:1000, New England Biolabs, Ipswich, MA, USA, #M0314) and cycloheximide (0.5 mg/mL, Sigma Aldrich, #C-4859). Input was precleared twice with protein A beads (10 μL per 100 μL sample/supernatant) to reduce non-specific binding. Protein A beads (10 μL, New England Biolabs, #S1425) were washed twice in coIP-solution and consecutively incubated for 1 h at 4 °C with an anti-synaptopodin antibody (rabbit anti-synaptopodin, Synaptic Systems, Göttingen, Germany, #163002). Beads were carefully washed twice to remove unbound antibodies and incubated for 1 h with the precleared supernatant (‘input’) at 4 °C with head-over-tail rotation. An 10 μL volume of the precleared supernatant was kept as input control. After incubation, beads were washed twice with coIP-solution and finally dissolved in RNA protection buffer (New England Biolabs).

In another round of experiments, hippocampal tissue from adult animals was used for co-immunoprecipitation experiments (antibody: monoclonal rabbit anti-synaptopodin, Abcam, #ab259976) with subsequent negative contrast electron microscopy. Beads (30 μL) were adsorbed on carbon-coated films stabilized by a copper grid and stained with 3% uranyl acetate (Serva, Heidelberg, Germany). Samples were immediately examined with a transmission electron microscope (Morgagni 268, 80 kV, FEI, Eindhoven, The Netherlands).

RNA was released from the beads through protein K lysis (Monarch^®^ Total RNA Miniprep Kit; New England Biolabs, #T2010S). After lysis, beads were removed from the solution in a magnetic rack and the RNA containing supernatant was transferred to fresh tubes. RNA isolation on coIP and input samples was then performed according to the manufacturer’s instructions. RNA content was determined using the Agilent RNA 6000 Pico Kit (Agilent, #5067-1513) with a 2100 Bioanalyzer (Agilent, #G2939BA). RNA Library preparations for Synpo-related transcriptome analysis were performed using the NEBNext^®^ Single Cell/Low Input RNA Library Prep Kit for Illumina^®^ (New England Biolabs, #E6420) according to the manufacturer’s instructions. We quantified the libraries using the NEBNext Library Quant Kit for Illumina (New England Biolabs, #E7630) based on the mean insert size provided by the Bioanalyzer. A 10 nM sequencing pool (120 μL in Tris-HCl, pH 8.5) was generated for sequencing on the NovaSeq6000 Sequencing platform (Illumina; service provided by CeGaT GmbH, Tübingen, Germany). We performed a paired-end sequencing with 150 bp read length. Data analysis was performed at the Galaxy platform (usegalaxy.eu; accession date: 01 August 2023). All files contained more than 10 M high-quality reads (after mapping to the reference genome; mm10) having at least a phred quality of 30 (>90% of total reads).

### 2.10. Regional mRNA Library Preparations and Transcriptome Analysis

RNA Library preparations for transcriptome analysis were performed using the NEBNext^®^ Single Cell/Low Input RNA Library Prep Kit for Illumina^®^ (New England Biolabs, #E6420) according to the manufacturer’s instructions. Briefly, isolation of the dentate gyrus and CA3 area from individual tissue cultures was performed using a scalpel. One isolated dentate gyrus or CA3 area, respectively, was transferred to 7.5 μL lysis buffer (supplemented with murine RNase inhibitor) and homogenized using a pestle. Samples were centrifuged for 30 s at 10,000× *g*, and 5 μL of supernatant was collected from individual samples and further processed. After cDNA synthesis, cDNA amplification was performed according to the manufacturer’s protocol with 12 PCR cycles. cDNA yield was consecutively analyzed using a High Sensitivity DNA assay on a Bioanalyzer instrument (Agilent). cDNA amount was adjusted to 10 ng for further downstream applications. After fragmentation and adaptor ligation, dual index primers (New England Biolabs, #E7600S) were ligated in a library amplification step using 10 PCR cycles. Libraries were finally cleaned up with 0.8X SPRI beads following a standard bead purification protocol. Library purity and size distribution was assessed with a High Sensitivity DNA assay on a Bioanalyzer instrument (Agilent). We quantified the libraries using the NEBNext Library Quant Kit for Illumina (New England Biolabs, #E7630) based on the mean insert size provided by the Bioanalyzer. A 10 nM sequencing pool (120 μL in Tris-HCl, pH 8.5) was generated for sequencing on the NovaSeq6000 Sequencing platform (Illumina; service provided by CeGaT GmbH, Tübingen, Germany). We performed a paired-end sequencing with 100 bp read length. Data analysis was performed at the Galaxy platform (usegalaxy.eu). All files contained more than 10 M high-quality reads (after mapping to the reference genome; mm10) having at least a phred quality of 30 (>90% of total reads).

### 2.11. Experimental Design and Statistical Analysis

Electrophysiological data were analyzed using pClamp 10.7 (Axon Instruments) and MiniAnalysis (version 6.0, Synaptosoft, Decatur, GA, USA) software. The fraction of action potentials (AP) followed by time-locked excitatory postsynaptic current responses was considered as synaptic response rate. uEPSC (unitary excitatory postsynaptic current) amplitude was assessed in uEPSCs from successfully transmitted action potentials as well as the mean amplitude of all successfully evoked postsynaptic responses. In order to ensure analysis of time-locked unitary responses, events were included if >5% of presynaptic pulses were successfully transmitted for each pulse, respectively. For individual pulses in pulse trains that caused ≤5% successfully transmitted events, response rate was set to ‘0′. sEPSC properties were analyzed using the automated template search tool for event detection. Since sEPSC kinetics might contain information on their synaptic origin (in CA3-PCs, large EPSCs have been identified to originate from hippocampal mossy fibers [29]), we performed a hierarchical analysis of sEPSCs based on their amplitude. All events of a recorded cell were sorted based on their peak amplitude. Events above the 50th percentile therefore correspond to all events above the cells’ median amplitude of events. Amplitude and half width data from sEPSC analysis have been plotted to visualize sEPSC kinetics, and linear fitting was employed to detect changes in parameter interdependencies. sEPSC area was reported as a function of amplitude and half width. Input resistance was calculated for the injection of −100 pA current at a time frame of 200 ms with a maximum distance to the initial hyperpolarization. Resting membrane potential was calculated as the mean baseline value. AP detection was performed using the input–output curve threshold search event detection tool, and the AP frequency was assessed by the number of APs detected during the respective current injection time.

Three-dimensional reconstructions of individual dendritic spines in electron micrograph image stacks were performed using the TrakEM2 tool in the Fiji software environment by 3 individual investigators blind to experimental conditions and hypothesis [30]. From one image stack, subfields were selected (proximal to soma: dentate gyrus (DG), 10–50 μm; CA3, 30–90 μm distance to soma; distal to soma: >100 μm distance to soma in both subfields) and all pre- and postsynaptic compartments of asymmetric synapses that were fully captured on image stacks were manually reconstructed. In order to analyze synaptic complexity, the images with the 3D-reconstructed synaptic compartments were used. The number of postsynaptic compartments and their postsynaptic densities were manually counted for each respective presynaptic compartment by an investigator blinded to experimental conditions.

Transmission electron microscopy images of mossy fiber synapses were saved as TIF-files and analyzed using the ImageSP Viewer software (http://e.informer.com/sys-prog.com, accessed on 1 August 2023, Sysprog, Minsk, Belarus). Ultrastructural features were analyzed by an investigator blind to experimental conditions in randomly selected mossy fiber terminals from electron micrographs of the stratum lucidum in non-lesioned and lesioned wildtype tissue cultures.

RNA sequencing data were uploaded to the Galaxy web platform (public server: usegalaxy.eu; [31,32,33]), and transcriptome analysis was performed using the Galaxy platform in accordance with the reference-based RNA-seq data analysis tutorial [34]. Adapter sequencing, low quality, and short reads were removed via the CUTADAPT tool. Reads were mapped using RNA STAR with the mm10 full reference genome (Mus musculus). The evidence-based annotation of the mouse genome (GRCm38), version M25 (Ensembl 100), served as gene model (GENCODE). For an initial assessment of gene expression, unstranded FEATURECOUNT analysis was performed from RNA STAR output. Only samples that contained >60% unique mapping reads (feature: “exon”) were considered for further analysis. Statistical evaluation was performed using DESeq2 with “pathway integrity” (ECL experiments) or “coIP” (synaptopodin-coIP experiments) as the primary factor that might affect gene expression. Genes with a low number of mean reads were excluded from further analysis. Genes were considered as differentially expressed or enriched if the adjusted *p*-value was <0.05. Data visualization was performed according to a modified version of a previously published workflow [35]. Further functional enrichment analyses were performed using g:Profiler (version e107_eg54_p17_bf42210) with g:SCS multiple testing correction method applying significance threshold of 0.05 [36]. Gene sets with 100–500 terms were considered for illustration. Heatmaps were generated based on z-scores of the normalized count table.

For transcriptome analysis following synaptopodin-coIP, DESeq2 analysis was performed to compare coIP and input samples. Differentially enriched genes in coIP (p_adj_ < 0.05 and log2(FC) > 0) were considered for further analyses. To correct for non-specific antibody binding, DESeq2 output (log2(FC)) from the same experimental procedure in synaptopodin-deficient tissue cultures was subtracted from log2(FC) of differentially enriched genes in wildtype cultures. Only those differentially expressed genes that showed >30% enrichment in FC after Synpo-KO correction were considered as significantly enriched.

Data were statistically analyzed using GraphPad Prism (Version 7.05 or 9.4.1, GraphPad software, Boston, MA, USA). For statistical comparison of two experimental groups, a Mann–Whitney test was employed. For statistical comparison of three experimental groups, a Kruskal–Wallis test was employed. In the graphs demonstrating volumes of the synaptic compartments, the box depicts 25–75 percentile, whiskers depict 10–90 percentile and the line indicates the median. Values outside this range were indicated by individual dots. Otherwise, values represent the mean ± standard error of the mean (s.e.m.). sEPSC amplitude/frequency plots and AP-frequency plots were statistically assessed via the repeated measure (RM) two-way ANOVA test with Sidak’s (two groups) multiple comparisons test. uEPSC amplitude values from individual cells were stacked in subcolumns, and the pulse number defined tabular rows (COLUMN factor: pathway integrity, genetic background; ROW factor: EPSC amplitude bin or current injection). *p*-values < 0.05 were considered statistically significant (* *p* < 0.05, ** *p* < 0.01, *** *p* < 0.001); results that did not yield significant differences are designated ‘ns’. Statistical differences in XY-plots were indicated in the legend of the figure panels (*) when detected through multiple comparisons, irrespective of their localization and the level of significance.

### 2.12. Data Availability

Source files and detailed statistical analyses are available from the Dryad data repository (https://doi.org/10.5061/dryad.x0k6djhpc). Sequencing data have been deposited in the Gene Expression Omnibus (GEO) repository (accession number: GSE216509). Original data are available from the corresponding authors upon reasonable request.

### 2.13. Digital Illustrations

Confocal image stacks were stored as TIF files. Figures were prepared using the ImageJ software package (https://imagej.nih.gov/ij/; accessed on 1 August 2023) and Photoshop graphics software (Version 24.4.1, Adobe, San Jose, CA, USA). Image brightness and contrast were adjusted.

## 3. Results

### 3.1. Entorhinal Cortex Lesion Leads to Excitatory Synaptic Strengthening in Both dGCs and CA3-PCs

Entorhinal cortex lesion was performed in mature tissue cultures to achieve a partial denervation of both dGCs and CA3-PCs (Figure 1A). Spontaneous excitatory postsynaptic currents (sEPSCs) were recorded from dGCs and CA3-PCs in non-lesioned control and lesioned tissue cultures (3 days post entorhinal cortex lesion, 3dp ECL) to assess denervation-induced effects on excitatory neurotransmission (Figure 1B). As previously described (e.g., [13,15]), partial denervation caused a strengthening of excitatory synapses in dGCs, i.e., a significant increase in both the mean sEPSC amplitude and half width as well as the mean sEPSC area (Figure 1C). Percentile analysis revealed that the lesion-induced increase in sEPSC amplitudes is evident when comparing the 50% biggest events in both groups, while no effect was observed when comparing smaller events (Figure 1D). Notably, a significant upregulation in the mean sEPSC frequency was observed after lesion (Figure 1E) that resulted from an increased abundance of high amplitude sEPSCs, as revealed by the amplitude/frequency plot (Figure 1F).

Next, we examined excitatory neurotransmission in CA3-PCs in non-lesioned and lesioned cultures (3dp ECL; Figure 1G–J). Again, a significant increase in both the mean sEPSC amplitude and area was observed in CA3-PCs from lesioned cultures, indicating a denervation-induced excitatory synaptic strengthening (Figure 1G). Notably, the mean sEPSC half width remained unchanged. In the sEPSC amplitude percentile analysis, excitatory synaptic strengthening seems to be restricted to the high-amplitude events (Figure 1H). Moreover, the mean sEPSC frequency remained unchanged upon ECL (Figure 1I), which resulted from an increase in the abundance of high-amplitude events at the expense of small-amplitude events (Figure 1J, RM two-way ANOVA followed by Sidak’s multiple comparisons test). We therefore conclude that ECL strengthens excitatory neurotransmission in both dGCs and CA3-PCs.

### 3.2. Partial Denervation Does Not Cause Changes in Intrinsic Cellular Properties

Passive and active membrane properties were assessed in both dGCs (Figure 2A,C,E) and CA3-PCs (Figure 2B,D,F) to test for adaptations of intrinsic cellular properties upon lesion. In dGCs no changes in passive membrane properties (Figure 2A), i.e., the resting membrane potential (RMP), and the input resistance as well as the input–output curve dynamics were observed (Figure 2C). Additionally, the action potential frequency of dGCs was similar in lesioned and non-lesioned cultures (Figure 2E). Of note, RMP assessment confirmed the mature state of the recorded dGCs (c.f. [37]).

Concerning CA3-PCs, neither the passive membrane properties (Figure 2D)—including input–output curve dynamics—nor the active properties (Figure 2F) were changed. Based on these results, we conclude that intrinsic cellular properties were stable following denervation in both tested cell types of the hippocampal network.

### 3.3. Entorhinal Cortex Lesion Induces Ultrastructural Changes in Dendritic Spines

To visualize lesion-induced effects on the ultrastructural level in a time-dependent manner, we performed 3D electron microscopy (based on SBF-SEM) in both the dentate gyrus (DG) and the CA3 subfields (Figure 3). Regions of interest were identified in the image stacks from proximal (DG, 10–50 μm; CA3, 30–90 μm distance to soma) and distal apical dendrites (>100 μm distance to soma in both subfields). Three-dimensional reconstruction was performed in randomly selected areas in the region of interest, and all synaptic compartments of asymmetric spine synapses that were fully included in the image stacks were analyzed in non-lesioned control cultures and 2 h (2 hp ECL) as well as 3 days (3 dp ECL) after entorhinal cortex lesion (Figure 3A,F). In the DG, proximal dendrites that received intrahippocampal inputs showed a reduction in postsynaptic volume, while the presynaptic compartment remained unchanged (Figure 3B). Interestingly, the complexity of synaptic organization, which was measured by the number of postsynapses at the very same presynaptic bouton, increased constantly over time (Figure 3C). Distal dendritic synapses showed a decline in both pre- and postsynaptic volumes upon lesion (Figure 3D), while synaptic complexity remained unchanged. In addition, ECL caused alterations in the surface area of postsynaptic densities in the DG (PSDs, Table 1): while the surface area of PSDs remained unchanged at proximal dendrites, it decreased significantly at distal dendrites after ECL.

In the CA3 subfield of proximal synapses, entorhinal cortex lesion caused an increase in presynaptic volumes 3 days post lesion, while the postsynaptic volume remained stable (Figure 3G). In contrast, distal dendritic synapses showed a decrease in postsynaptic volumes, while the presynaptic compartment remained unchanged (Figure 3I). Here, synaptic complexity was unchanged in both proximal and distal dendritic compartments (Figure 3H,J). Moreover, the surface area of PSDs increased significantly at proximal synapses in the CA3 subfield 3 days after lesion (Table 1). At the same time, at the distal synapses there was a significant decrease in PSD surface area.

### 3.4. Entorhinal Cortex Lesion Induces Changes in Synaptic Transmission at Hippocampal Mossy Fiber Synapses

Hippocampal mossy fiber synapses have been linked to learning processes and memory formation [38,39]. They are characterized by their generation of large postsynaptic currents upon activation [29]. Given that our results hint towards a preferential lesion-related plasticity induction in readily strong synapses of CA3-PCs, we next assessed the features of synaptic transmission at hippocampal mossy fiber synapses.

Therefore, we performed paired whole-cell patch-clamp recordings of individual dGCs and CA3-PCs in organotypic tissue cultures (Figure 4A). To test for synaptic connections, five consecutive action potentials at 20 Hz frequency were elicited in dGCs (Figure 4B). Unitary excitatory postsynaptic currents (uEPSCs) were monitored in CA3-PCs in non-lesioned and lesioned tissue cultures. We found that a train of presynaptic action potentials in connected neurons induced a robust facilitation of postsynaptic responses in both postsynaptic response rate and uEPSC amplitude under baseline conditions (Figure 4B–D). Notably, the response rate to the first presynaptic action potential, which is the number of presynaptic action potentials that generate a postsynaptic response, was low in non-lesioned cultures (~3%; Figure 4C). In lesioned cultures, however, a significant increase in the synaptic reliability for the first two pulses was observed. For the last three pulses in one train of stimulation, averaged synaptic response rate levels became similar (Figure 4C). Moreover, the uEPSC amplitude was significantly increased in response to the individual pulses 3 days after ECL (Figure 4D). The dGC-to-CA3-PC paired recordings in the steady-state environment of organotypic tissue cultures therefore provided evidence that synaptic transmission at hippocampal mossy fiber synapses is a target for lesion-induced remodeling of hippocampal networks.

### 3.5. Synaptopodin and the Spine Apparatus Organelle Are Associated with Dendritic Ribosomes and Synapse-Related mRNAs

Hippocampal mossy fiber synapses show a complex organization of their pre- and postsynaptic sites. Among these features, a regular occurrence of the spine apparatus organelle has been reported at the postsynaptic site [5]. Using transmission electron microscopy, a close correlation between ribosomes and the spine apparatus organelle was readily revealed (Figure 5A). Therefore, we quantified the presence of ribosomes in randomly selected dendritic spines that either contain or lack the spine apparatus organelle at proximal apical dendritic segments in transmission electron micrographs of the CA3 subfield (Figure 5B). We found that approx. 50% of dendritic spine cross-sections that lacked a spine apparatus contained ribosomes. When the spine apparatus organelle was present in dendritic spine cross sections, however, approx. 90% contained ribosomes. To strengthen the association between synaptopodin/the spine apparatus organelle and ribosomes, we performed synaptopodin co-immunoprecipitation experiments of hippocampal lysates from adult animals followed by negative contrast electron microscopy (Figure 5C). Here, we found ribosomes organized along filamentous structures. Since these findings suggested an association between synaptopodin/the spine apparatus organelle and ribosomes, which might contribute to plasticity-related local protein synthesis, we performed a transcriptome analysis of Synpo-coIP enriched mRNA (Figure 5D, Appendix A). After correction of unspecific antibody pull-downs by using synaptopodin-KO tissue lysates, a considerable number of mRNAs was enriched after Synpo-coIP (violet dots; 1288 mRNAs). Subsequently, a gene set enrichment analysis (g:Profiler) was performed to elucidate the functional relevance of synaptopodin-associated mRNAs (Figure 5E). Among others, an enrichment in synapse-related gene sets was identified. Therefore, these data hint towards a direct or indirect functional association between synaptopodin/the spine apparatus organelle and ribosomes, which might contribute to lesion-induced synaptic plasticity through local protein synthesis.

### 3.6. Ultrastructural Reorganization of the Spine Apparatus Organelle

Previous studies have demonstrated that the ultrastructural organization of the spine apparatus organelle might represent a target for plasticity [8,19]. To study lesion-induced effects on the ultrastructural organization of the spine apparatus organelle, we performed transmission electron microscopy of mossy fiber synapses in non-lesioned and lesioned (3 dp ECL) entorhino-hippocampal tissue cultures (Figure 5F; c.f., [8]). No differences in the structural organization of pre- and postsynaptic sites could be detected between the groups (presynapse: mossy fiber perimeter: 11.61 ± 0.4 μm (control; n = 120) vs. 10.88 ± 0.35 μm (3 dp ECL; n = 90); postsynapse: number of PSDs: 4.91 ± 0.33 (control; n = 120) vs. 4.6 ± 0.27 (3 dp ECL; n = 90), length of PSDs: 201 nm ± 4.35 (control; n = 589) vs. 201.7 ± 5.42 nm (3 dp ECL; n = 414); values indicate mean ± sem, Mann–Whitney test). Nevertheless, a substantial reorganization of the spine apparatus organelle was evident in lesioned cultures 3 days after injury compared to non-lesioned controls (Figure 5G,H). We found an increased number of dense plates in spine apparatus organelles from lesioned cultures (Figure 5G). At the same time the length of individual dense plates was significantly reduced in lesioned cultures (Figure 5H). Therefore, we conclude that the spine apparatus organelle represents a target for lesion-induced changes at hippocampal mossy fiber synapses.

### 3.7. Synaptopodin-Deficient Tissue Cultures Show Deficits in Lesion-Induced Synaptic Plasticity upon Partial Denervation

Based on our ultrastructural analysis, we hypothesized that the spine apparatus organelle might contribute to lesion-induced synaptic plasticity at hippocampal mossy fiber synapses. To gain further insights concerning the role of the spine apparatus organelle, we performed experiments in synaptopodin-deficient tissue preparations, which lack the spine apparatus organelle (Figure 6 [20]).

Synaptopodin-deficient dGCs showed a significant increase in both the mean sEPSC amplitude and area as well as the mean sEPSC half width 3 days following ECL (Figure 6A). In contrast, no changes in the mean sEPSC frequency, which have been observed in wildtype (Synpo^+/+^) cultures, were detected (Figure 6B). Moreover, passive membrane properties remained stable upon lesion in synaptopodin-deficient tissue preparations (Figure 6C).

In CA3-PCs, no significant changes in the sEPSC event analysis (Figure 6D) or the mean sEPSC frequency (Figure 6E) were observed upon lesion. Again, passive membrane properties remained unchanged (Figure 6F). Thus, our findings highlight altered lesion-induced synaptic adaptations for both dGCs and CA3-PCs in synaptopodin-deficient tissue cultures.

We extended these findings with dGC-to-CA3-PC paired whole-cell patch-clamp recordings in synaptopodin-deficient preparations (Figure 6G). In contrast to wildtype cultures, no significant differences in the synaptic reliability could be detected 3 days after ECL (Figure 6H). We found a significant reduction in the pulse-related uEPSC amplitude of successfully transmitted events in lesioned synaptopodin-deficient preparations (Figure 6I). Notably, excitatory neurotransmission seemed facilitated under baseline conditions as compared to wildtype cultures as indicated by significant changes in sEPSC half width, area and amplitude/frequency plots (Figure 7A,B). Moreover, changes in uEPSCs at mossy fiber synapses were evident in paired recordings (Figure 7C,D). Hence, synaptopodin-deficient mossy fiber synapses responded with a synaptic depression rather than strengthening after the lesion. Taken together, these findings demonstrate the functional relevance for the presence of synaptopodin in regulating lesion-induced plasticity at hippocampal mossy fiber synapses.

### 3.8. Entorhinal Cortex Lesion Causes Synaptopodin-Dependent Changes in the Region-Specific Hippocampal Transcriptome

To further explore synaptopodin-dependent changes in hippocampal subfields upon lesion, we performed transcriptome analyses of both the DG and the CA3 region in non-lesion and lesioned wildtype (Synpo^+/+^) and synaptopodin-deficient (Synpo^−/−^) cultures (Figure 8). The expression analysis revealed a considerable number of genes that are differentially expressed in both hippocampal regions from wildtype animals (Figure 8A,B). In contrast, synaptopodin-deficient tissue culture preparations showed deficits in the lesion-induced adjustment of gene expression in the CA3 region (Figure 8A). Moreover, wildtype and synaptopodin-deficient tissue preparations showed a genotype dependent gene expression adjustment upon lesion in both regions, although a common set of regulated genes was identified (Figure 8B; for detailed tabular results see Appendix A). We therefore conclude that lesion-induced transcriptomic changes might depend on synaptopodin.

Unbiased gene set enrichment analysis for differentially expressed genes (gene ontology–molecular function; GO:MF) from wildtype CA3 samples revealed that ECL caused changes in gene expression related to ubiquitin- and GTP-related signaling (Figure 8C). Additionally, significant enrichments in vesicle-, glia- and synapse-related gene sets were evident (gene ontology–cellular compartment; GO:CC). In order to learn more about lesion-induced differential expression of synaptic gene sets in the CA3 region (“synapse”; GOCC: 0045202), differentially expressed gene lists were filtered according to the content of the respective gene ontology (Figure 8D). Here, 153 differentially expressed synaptic genes were identified.

The same analyses were performed for wildtype DG samples (Figure 8E,F). In addition to changes in ubiquitin- and GTP-related signaling genes, enrichments were seen for actin and transcription factor binding related genes (gene ontology–molecular function; GO:MF, Figure 8E). Moreover, endosome and mitochondria related gene sets were enriched (gene ontology–cellular compartment; GO:CC). Again, 160 differentially expressed synaptic genes were identified in DG samples upon lesion (Figure 8F).

We conclude that the entorhinal cortex lesion caused regional transcriptome changes that might contribute to the implementation of lesion-induced synaptic plasticity.

## 4. Discussion

Denervation is a common phenomenon in numerous medical conditions affecting the central nervous system, which can trigger lesion-induced reorganization in neural networks [15,40,41,42,43,44]. In line with previous findings that employed proximity proteomics of synaptopodin [45], we identified an association between synaptopodin and ribosomes. Using synaptopodin co-immunoprecipitation experiments, we report a synaptopodin-related transcriptome, which might contribute to lesion-induced plasticity. Together with findings that indicate interactions between synaptopodin, ribosomal subunits and ER-related proteins [45], we suggest a role for synaptopodin as local regulator of synaptic plasticity at hippocampal mossy fiber synapses.

To study lesion-induced changes in synaptic transmission, we used the in vitro model of ECL in organotypic entorhino-hippocampal tissue cultures. After in vitro maturation, tissue organization reaches a steady-state that is characterized by an in vivo-like ultrastructural synaptic architecture [46,47]. In this context, recent work has demonstrated that organotypic hippocampal tissue cultures have intact mossy fiber pathways and synapses [6]. Moreover, the excitatory cortical input—the perforant path—terminates at distal apical dendrites of dGCs and pyramidal neurons of the ammonic subregions [48,49,50,51]. However, projections from other cortical and subcortical regions and other environmental factors are missing in in vitro systems that might influence synaptic transmission and plasticity in the hippocampus [52,53].

Our results demonstrate lesion-induced excitatory synaptic strengthening in both dGCs and CA3-PCs, which is accompanied by changes in the region-specific transcriptome. Moreover, here we used paired whole-cell patch-clamp recordings to assess synaptic transmission at individual hippocampal mossy fiber synapses—the monosynaptic excitatory contacts between dGCs and CA3-PCs—in a steady-state environment. In this experimental approach, we identified denervation-induced adaptations at hippocampal mossy fiber synapses. In addition, ultrastructural changes in the volume of synapses as well as in the spine apparatus organelle, which is well established in the regulation of synaptic plasticity [8,19], were observed after ECL. Since adaptations on both synaptic and transcriptomic levels were altered in synaptopodin-deficient tissue cultures, we assert a crucial regulatory role for synaptopodin and the spine apparatus organelle in orchestrating lesion-induced plasticity.

Lesion-induced excitatory synaptic strengthening might aim at compensating for the loss of input. The compensatory adjustment of excitatory synaptic strength, which aims at re-achieving distinct physiological setpoints in neural networks, has been summarized as homeostatic synaptic plasticity [54]. In this study, we did not observe changes in dGC firing rates upon lesion that might be restored by synaptic adaptations in a homeostatic fashion. However, the stability and adjustment of excitatory neurotransmission after lesion suggest the recruitment of homeostatic mechanisms. Since our 3D reconstruction of synapses revealed the rapid implementation of changes in synapse structure and complexity, it is intriguing to theorize that compensatory plasticity at remaining intra-hippocampal pathways might operate on different time scales to rapidly stabilize neuronal function upon denervation.

The dentate gyrus/CA3 neural network is an integral component of the trisynaptic hippocampal pathway. The hippocampal mossy fiber pathway forms large en passant synapses and connects one individual dGC to a limited number of CA3-PCs [55]. Both regions receive cortical input—the perforant path—from entorhinal layer II stellate cells that can induce, promote and modulate information processing in hippocampal subregions. This pathway is known to be implicated in various diseases; among them traumatic, inflammatory and neurodegenerative conditions like Alzheimer’s disease (AD; [56,57]). Additionally, the hippocampal mossy fiber synapse has been linked to memory acquisition and retrieval [58,59]. Specifically, activity-dependent structural and functional changes in synaptic features can encode input patterns for a prolonged period of time [11,39]. Again, alterations in hippocampal mossy fiber transmission have been reported in animal models of Alzheimer’s disease (AD) and aging-related cognitive decline [60,61]. Nevertheless, it remains unclear if the lesion-induced synaptic response deteriorates cognitive performance by affecting information processing in neural networks.

Recently, characteristic features of transmission and stabilizing factors at hippocampal mossy fiber synapses have been revealed [62,63,64,65]. Upon presynaptic activity, a detonating behavior can be observed, in which a limited number of—but not single—action potentials can elicit a suprathreshold postsynaptic depolarization [66]. Although this phenomenon has been extensively described in mossy fiber bouton-CA3-PC paired recordings [67], we were not able to reveal detonating characteristics in dGC-CA3-PC pairs. Here, differences in the experimental model and methodology can account for the divergent observation of detonator characteristics. In line with previous reports, however, a filtering of single action potentials with a subsequent robust facilitation of synaptic transmission at this synapse was evident [68,69]. Our results demonstrate that the partial loss of input following lesion can influence both synaptic reliability and potency, which has been observed for other plasticity-inducing stimuli at this synapse [69]. We suggest that lesion-induced modulation of filtering/facilitation properties might interact with network phenomena, such as pattern separation and completion along the dentate gyrus/CA3-axis [70,71,72]. Moreover, the presence of synaptopodin may interfere with this network process under baseline and plasticity-inducing conditions accounting for the previously reported cognitive deficits in synaptopodin-deficient mice [20] and the synaptopodin-related cognitive decline during human aging [10].

Of note, it is an established view that mossy fiber synaptic plasticity is predominantly mediated by presynaptic processes [39,73,74]. Nevertheless, recent work has suggested that the postsynaptic site shapes the transmission and plasticity through transsynaptic mechanisms [63]. In this study, we report that the spine apparatus organelle regulates denervation-induced synaptic plasticity at this synapse. The spine apparatus organelle, which is lacking in synaptopodin-deficient tissue [20], was previously linked to learning deficits in adult mice and alterations in synaptic plasticity in various research models [75]. Of note, synaptopodin protein expression and spine apparatus formation start postnatally [76]. Therefore, synaptopodin-mediated plasticity regulation in mature systems seems to have a substantial contribution to the observed phenotypes. Since its discovery, however, the function of the spine apparatus organelle and the mechanisms through which it orchestrates synaptic plasticity have remained enigmatic. By using electron microscopy and co-immunoprecipitation experiments, we demonstrated here that synaptopodin and the spine apparatus organelle are related to ribosomes. However, it remains unclear if these are direct or indirect interactions, and co-immunoprecipitation might target mRNA transport/storage granules. Given the lesion-induced ultrastructural remodeling of the spine apparatus organelle, it is interesting to speculate whether these changes might orchestrate the previously reported local protein synthesis in close proximity to synaptopodin clusters [23,77]. Since (1) lesion-induced synaptic plasticity is altered in synaptopodin-deficient tissue preparations and (2) alterations in synaptopodin expression have been linked to cognitive decline [10], we hypothesize that synaptopodin-dependent plasticity might stabilize cognitive performance by increasing synaptic resilience [78].

In conclusion, our results demonstrate a role for synaptopodin in regulating synaptic transmission and lesion-induced plasticity at the hippocampal mossy fiber synapse in organotypic entorhino-hippocampal tissue cultures. Our data indicate a relationship between ribosomes and the spine apparatus organelle, which suggests that synaptopodin and the spine apparatus organelle might be involved in synapse related local protein synthesis.

## Figures and Tables

**Figure 1 cells-13-00114-f001:**
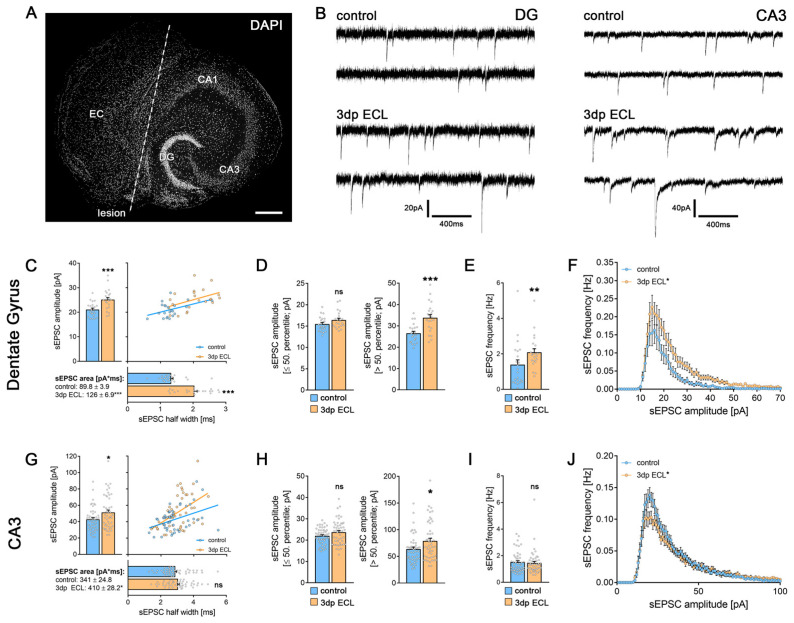
Entorhinal cortex lesion caused synaptic strengthening in both dGCs and CA3-PCs. (**A**) Entorhinal cortex lesion (ECL) in organotypic entorhino-hippocampal tissue cultures was performed by dissecting the perforant path (white line, scale bar 300 μm). Cytoarchitecture was visualized via DAPI nuclear staining. (**B**) Spontaneous excitatory postsynaptic currents (sEPSCs) were recorded in both dGCs and CA3-PCs in non-lesioned and lesioned cultures (3 days post lesion, 3 dp ECL). (**C**,**D**) Spontaneous EPSC analysis in dGCs revealed a lesion-induced excitatory synaptic strengthening in the mean sEPSC amplitude, half width and area ((**C**); n_control_ = 22 cells; n_3dp ECL_ = 22 cells; Mann–Whitney test; Linear regression fit for XY-plot analysis). Percentile analysis of sEPSC amplitudes (**D**) demonstrated that the excitatory synaptic strengthening was evident in the 50% largest events (Mann–Whitney test). (**E**,**F**) The frequency of sEPSCs was significantly increased 3 days after ECL ((**E**); Mann–Whitney test). Moreover, amplitude/frequency-analysis shows that the abundance of larger sEPSC amplitudes was higher in lesioned cultures when compared to non-lesioned controls ((**F**); RM-two-way ANOVA followed by Sidak’s multiple comparisons test). (**G**,**H**) Spontaneous EPSC analysis in CA3-PCs revealed a lesion-induced increase in mean sEPSC amplitude, whereas half width and area remained unchanged ((**G**); n_control_ = 48 cells; n_3dp ECL_ = 49 cells; Mann–Whitney test; Linear regression fit for XY-plot analysis). Percentile analysis of sEPSC amplitudes (**H**) demonstrated that the excitatory synaptic strengthening was only visible in the 50% largest events (Mann–Whitney test). (**I**,**J**) No lesion-induced changes in mean sEPSC frequency were detectable in CA3-PCs. Amplitude/frequency-analysis showed a reduced number of small events in CA3-PCs 3 days after ECL (Mann–Whitney test for column statistics and RM-two-way ANOVA followed by Sidak’s multiple comparisons test for XY-plot analysis). Individual data points are indicated by colored dots. Values represent mean ± s.e.m (* *p* < 0.05, ** *p* < 0.01, *** *p* < 0.001, ns—non-significant difference).

**Figure 2 cells-13-00114-f002:**
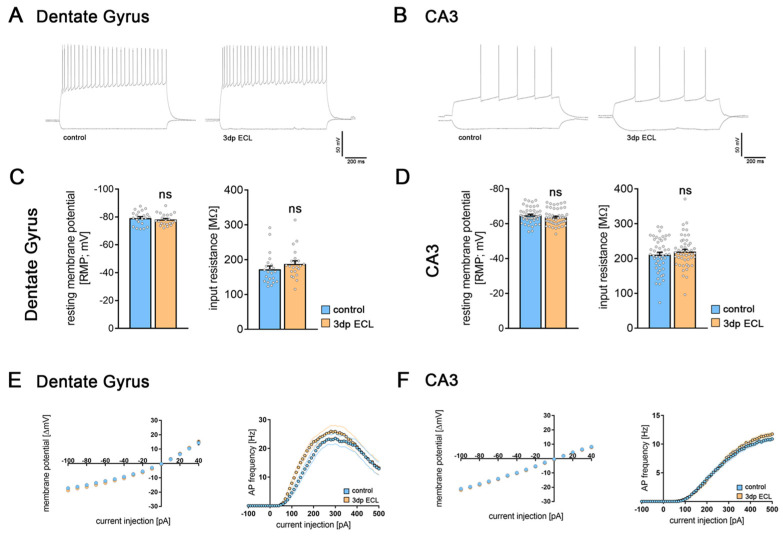
Entorhinal cortex lesion did not influence active or passive membrane properties. (**A**,**B**) Sample traces of dGCs (**A**) and CA3-PCs (**B**) in current clamp mode at a current injection of −100 pA (lower trace) and +300 pA (upper trace) in control cultures and 3 days post ECL. (**C**,**E**) Both resting membrane potential and input resistance (**C**) as well as the input–output curve for the first 15 sweeps of current injection (**E**) in dGCs were unaffected by ECL (n_control_ = 22 cells; n_3dp ECL_ = 22 cells; Mann–Whitney test and RM two-way ANOVA followed by Sidak’s multiple comparisons test for XY-plot analysis). (**E**) Action potential frequency upon increasing current injection in dGCs was similar in non-lesioned and lesioned cultures (RM two-way ANOVA followed by Sidak’s multiple comparisons test). (**D**,**F**) Both resting membrane potential and input resistance (**D**) as well as the input–output curve for the first 15 sweeps of current injection (**F**) in CA3-PCs were unaffected by the ECL (n_control_ = 48 cells; n_3dp ECL_ = 49 cells; Mann–Whitney test and RM two-way ANOVA followed by Sidak’s multiple comparisons test for XY-plot analysis). (**F**) Action potential frequency upon increasing current injection in CA3-PCs was similar in non-lesioned and lesioned cultures (RM two-way ANOVA followed by Sidak’s multiple comparisons test). Individual data points are indicated by colored dots. Values represent mean ± s.e.m (ns—non-significant difference).

**Figure 3 cells-13-00114-f003:**
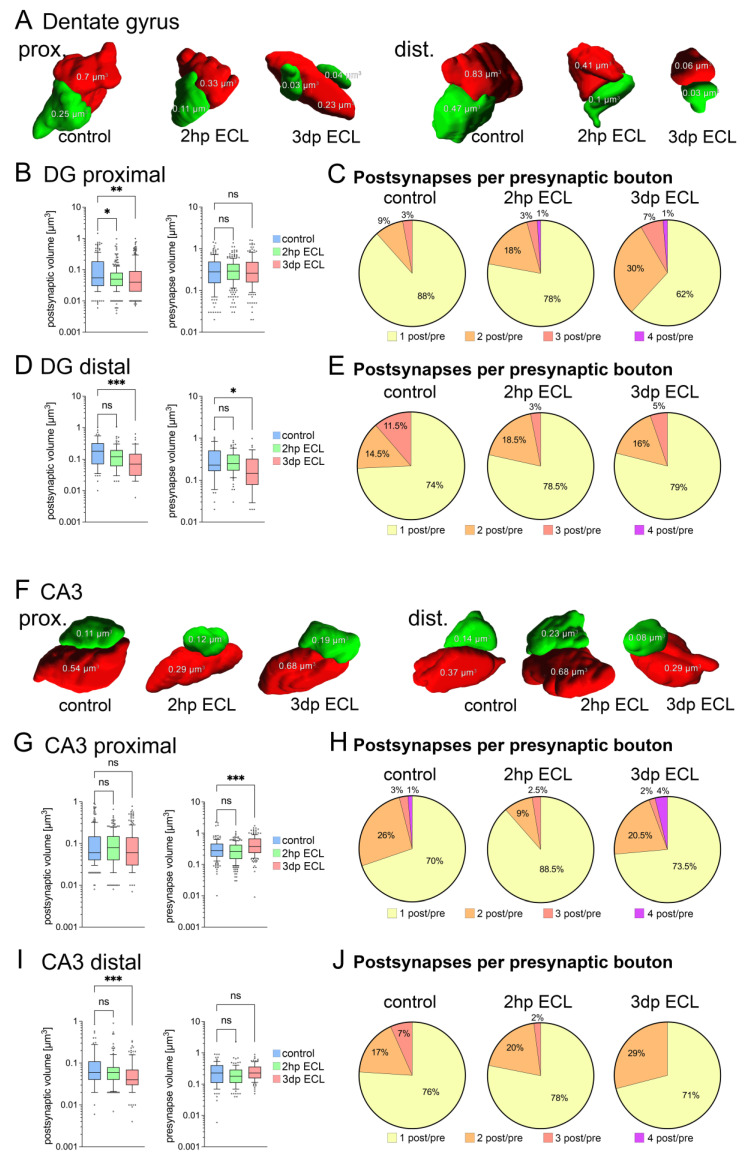
Entorhinal cortex lesion induced ultrastructural synaptic changes in DG and CA3 subfields. (**A**) Three-dimensional reconstructions of synapses (based on SBF-SEM) onto proximal and distal apical dendrites in the dentate gyrus (postsynaptic compartment, green; presynaptic compartment, red). (**B**) Analysis of both post- and presynaptic volumes from 3D reconstructions in proximal DG areas. A decrease in postsynaptic volume was detected after lesion while presynaptic volume remained stable (postsynaptic compartments: n_control_ = 170 postsynapses, n_2hp ECL_ = 286 postsynapses, n_3dp ECL_ = 223 postsynapses; presynaptic compartments: n_control_ = 163 presynapses, n_2hp ECL_ = 243 presynapses, n_3dp ECL_ = 158 presynapses from one representative tissue culture each; Kruskal–Wallis test followed by Dunn’s post hoc correction). (**C**) Analysis of synaptic complexity, which is the number of postsynaptic sites per presynaptic bouton, at proximal synapses in DG. Notably, synaptic complexity increases following entorhinal cortex lesion. (**D**) Analysis of both post- and presynaptic volumes in distal DG areas. A decrease in postsynaptic and presynaptic volumes were detected (postsynaptic compartments: n_control_ = 54 postsynapses, n_2hp ECL_ = 82 postsynapses, n_3dp ECL_ = 48 postsynapses; presynaptic compartments: n_control_ = 49 presynapses, n_2hp ECL_ = 74 presynapses, n_3dp ECL_ = 38 presynapses from one representative tissue culture each; Kruskal–Wallis test followed by Dunn’s post hoc correction). (**E**) Synaptic complexity at distal synapses in DG remained unchanged following lesion. (**F**) Three-dimensional reconstructions of synapses (based on SBF-SEM) onto proximal and distal apical dendrites in the CA3 region (postsynaptic compartment, green; presynaptic compartment, red). (**G**) Analysis of both post- and presynaptic volumes in proximal CA3 areas. An increase in presynaptic volume was detected while postsynaptic volumes remained unchanged (postsynaptic compartments: n_control_ = 208 postsynapses, n_2hp ECL_ = 224 postsynapses, n_3dp ECL_ = 219 postsynapses; presynaptic compartments: n_control_ = 159 presynapses, n_2hp ECL_ = 202 presynapses, n_3dp ECL_ = 168 presynapses from one representative tissue culture each; Kruskal–Wallis test followed by Dunn’s post hoc correction). (**H**) Synaptic complexity at proximal synapses in CA3 remained widely unchanged following lesion. (**I**) Analysis of both post- and presynaptic volumes in distal CA3 areas. A decrease in postsynaptic volume was evident while presynaptic volumes remained unchanged (postsynaptic compartments: n_control_ = 99 postsynapses, n_2hp ECL_ = 120 postsynapses, n_3dp ECL_ = 120 postsynapses; presynaptic compartments: n_control_ = 77 presynapses, n_2hp ECL_ = 98 presynapses, n_3dp ECL_ = 93 presynapses from one representative tissue culture each; Kruskal–Wallis test followed by Dunn’s post hoc correction). (**J**) Despite a trend towards reduced complexity, distal synapses in CA3 remained widely unchanged following entorhinal cortex lesion. Graphs depict box plots (box: 25–75 percentile; whiskers: 10–90 percentile; line indicates median), values outside this range were indicated by individual dots (* *p* < 0.05, ** *p* < 0.01, *** *p* < 0.001, ns—non-significant difference).

**Figure 4 cells-13-00114-f004:**
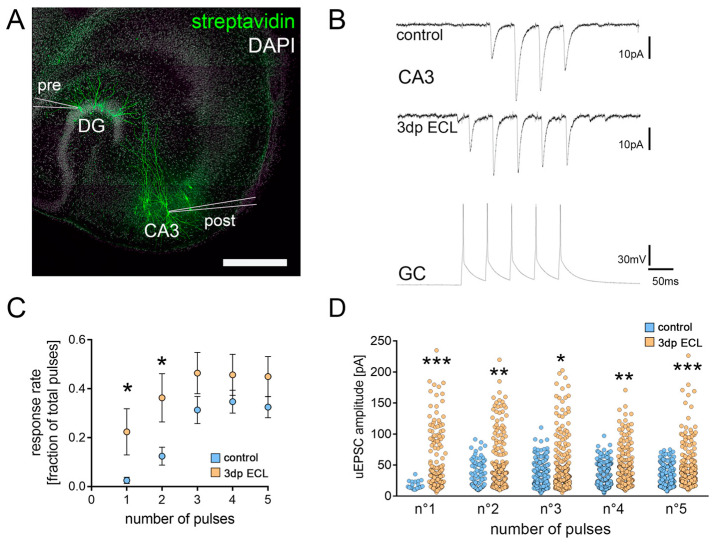
Lesion-induced loss of the cortico-hippocampal projection changed transmission at hippocampal mossy fiber synapses. (**A**) Post hoc visualization of patched neurons in organotypic tissue cultures after paired electrophysiological recordings between individual presynaptic (pre) GCs of the dentate gyrus (DG) and postsynaptic (post) CA3-PCs to analyze synaptic transmission features at hippocampal mossy fiber synapses (scale bar 300 μm). (**B**) Five consecutive action potentials were induced in dGCs at 20 Hz (bottom trace) to assess postsynaptic responses in CA3-PCs in lesioned and non-lesioned cultures (upper traces). (**C**) The response rate, i.e., the fraction of presynaptic pulses that elicited a postsynaptic response in a connected pair of individual neurons, was significantly increased for the first two pulses in lesioned cultures when compared to non-lesioned controls (n_control_ = 10; n_3dp ECL_ = 10; Mann–Whitney test for the respective pulse numbers). (**D**) In successfully elicited unitary EPSCs (uEPSCs), a significant increase in uEPSC amplitude was detectable (n°1_control_ = 20 events, n°1_3dp ECL_ = 187 events; n°2_control_ = 115 events, n°2_3dp ECL_ = 304 events; n°3_control_ = 292 events, n°3_3dp ECL_ = 394 events (one data point outside the axis limits); n°4_control_ = 326 events, n°4_3dp ECL_ = 385 events; n°5_control_ = 307 events, n°5_3dp ECL_ = 384 events; Mann–Whitney test for the respective pulse numbers). Individual data points are indicated by colored dots. Values represent mean ± s.e.m (* *p* < 0.05, ** *p* < 0.01, *** *p* < 0.001).

**Figure 5 cells-13-00114-f005:**
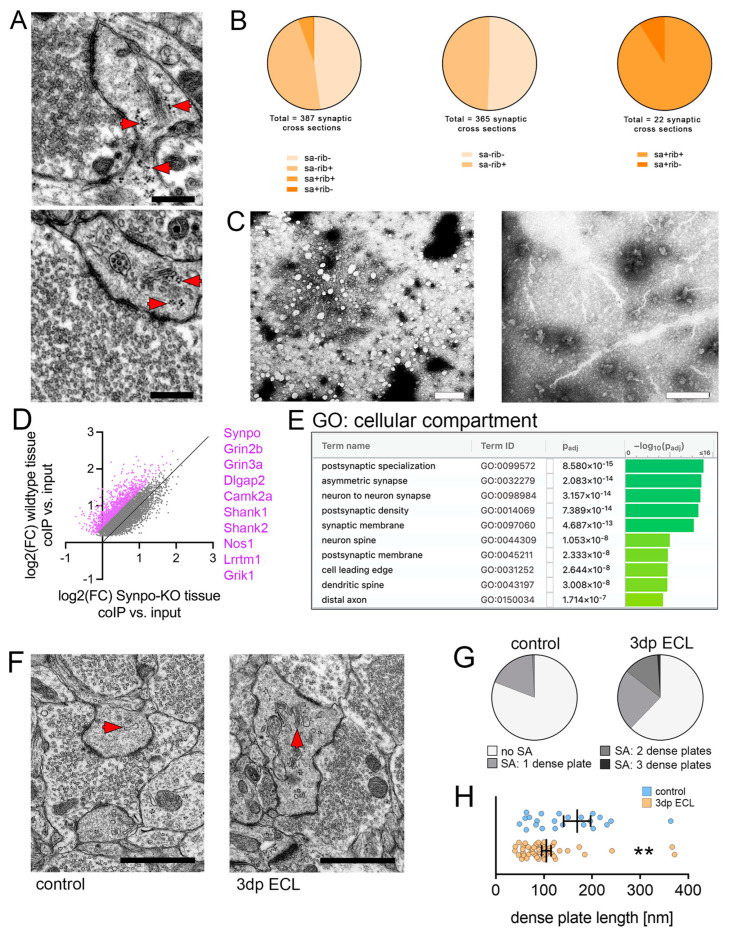
The spine apparatus organelle assembled ribosomes and excitatory-synapse-related mRNAs. (**A**) Transmission electron microscopy (TEM) images demonstrate a close association of ribosomes (arrowheads) and the spine apparatus organelle. Scale bar upper image, 300 nm; scale bar lower image, 400 nm. (**B**) Assessment of ribosomes in cross-sections of dendritic spines depending on the presence of the spine apparatus organelle (SA). In spine cross-sections lacking a spine apparatus organelle, approx. 50% of spines contained ribosomes. Notably, approx. 90% of the spines containing the spine apparatus organelle also contained ribosomes. (**C**) Negative contrast electron microscopy images from beads used from synaptopodin co-immunoprecipitation in hippocampal tissue from adult animals. Micrographs illustrate the immunoprecipitation of ribosomes and ribosomal subunits (white non-contrasted areas), which are organized along microfilaments suggesting an association between synaptopodin and ribosomes. Scale bars, 100 nm. (**D**) Analysis of differential mRNA abundance in Synpo-coIP RNA samples compared to their respective input samples to reveal the synaptopodin related transcriptome (Appendix A). A log2(FC) value > 0 indicates an mRNA enrichment in Synpo-coIP samples. To rule out the unspecific pull-down of targets, the same experimental protocol was applied to synaptopodin^−/−^ tissue lysates. Log2(FC) values were plotted for differentially enriched genes (p_adj._ < 0.05) from wildtype tissue cultures and specific enrichment was assumed if FC increased more than 30% (violet dots, n_wildtype_ = 5 samples and n_Synpo_^−/−^ = 2 samples input and Synpo-coIP, respectively; 5–6 tissue cultures pooled from one filter per sample; DESeq2-analysis). (**E**) A gene set enrichment analysis for a cellular compartment gene ontology was performed for significantly enriched genes in Synpo-coIP samples and visualized by the g:Profiler web application. Notably, synaptic gene sets were enriched. (**F**) Transmission electron microscopy was used to visualize ultrastructural properties of mossy fiber synapses in lesioned and non-lesioned control cultures (scale bar, 1 μm). Spine apparatus organelles are marked by red arrowheads. (**G**) Upon lesion, the abundance of spine apparatus organelles (SA) and their number of dense plates were significantly increased in the postsynaptic compartment of mossy fiber synapses (n_control_ = 120 (no SA = 97 postsynapses, 1 dense plate = 22 postsynapses, 2 dense plates = 1 postsynapse, 3 dense plates = 0 postsynapses); n_3dp ECL_ = 90 (no SA = 56 postsynapses, 1 dense plate = 21 postsynapses, 2 dense plates = 12 postsynapses, 3 dense plates = 1 postsynapses)). (**H**) The length of the respective dense plates significantly decreased 3 days after ECL (n_control_ = 24, one data point outside the axis limits; n_3dp ECL_ = 48; Mann–Whitney test). Individual data points are indicated by colored dots. Values represent mean ± s.e.m (** *p* < 0.01).

**Figure 6 cells-13-00114-f006:**
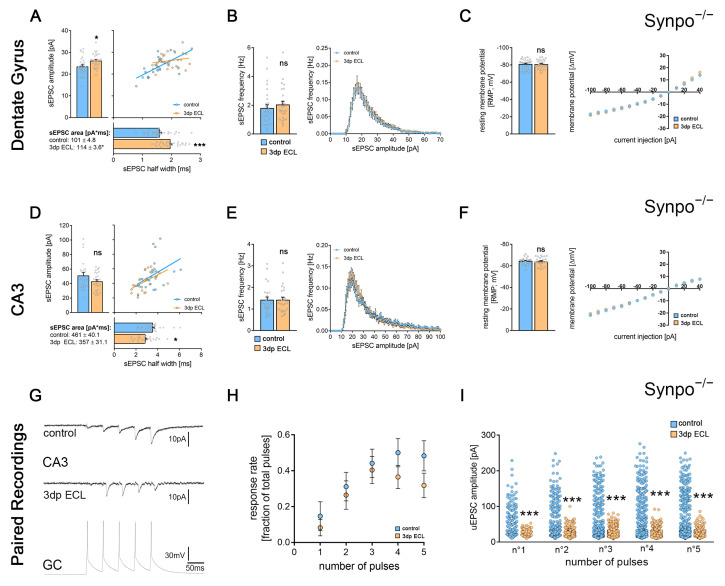
Lesion-induced synaptic plasticity at hippocampal mossy fiber synapses was impaired in synaptopodin-deficient tissue cultures. (**A**,**B**) sEPSCs were recorded from dGCs in synaptopodin-deficient tissue cultures (Synpo^−/−^). Here, the mean sEPSC amplitude, half width and area significantly increased 3 days after the ECL ((**A**); n_control_ = 29; n_3dp ECL_ = 26; Mann–Whitney test; Linear regression fit for XY-plot analysis). In contrast to wildtype cultures, no changes in the mean sEPSC frequency or the sEPSC amplitude/frequency-distribution were observed ((**B**); Mann–Whitney test for column statistics and RM-two-way ANOVA followed by Sidak’s multiple comparisons test for XY-plot analysis). (**C**) No changes in passive membrane properties of dGCs were detected in synaptopodin-deficient cultures upon lesion (Mann–Whitney test for column statistics and RM-two-way ANOVA followed by Sidak’s multiple comparisons test for XY-plot analysis). (**D**,**E**) The mean sEPSC amplitude in CA3-PCs was not significantly different in lesioned and non-lesioned control cultures. Noteworthy, the mean sEPSC half width significantly decreased in lesioned cultures ((**D**); n_control_ = 25; n_3dp ECL_ = 25; Mann–Whitney test; linear regression fit for XY-plot analysis). No lesion-induced changes in the mean sEPSC frequency or the amplitude/frequency-interdependence were detectable ((**E**); Mann–Whitney test for column statistics and RM-two-way ANOVA followed by Sidak’s multiple comparisons test for XY-plot analysis). (**F**) No changes in passive membrane properties of CA3-PCs were detected in synaptopodin-deficient cultures upon lesion (Mann–Whitney test for column statistics and RM-two-way ANOVA followed by Sidak’s multiple comparisons test for XY-plot analysis). (**G**,**H**) Paired whole-cell patch-clamp recordings were carried out between individual dGCs and CA3-PCs in lesioned and non-lesioned control cultures prepared from synaptopodin-deficient animals (**G**). Upon injection of 5 current pulses in the presynaptic granule cells, no changes in the pulse-number-dependent response rate of connected mossy fiber synapses were detectable after the ECL (H; n_control_ = 12; n_3dp ECL_ = 10; Mann–Whitney test for the respective pulse numbers). (**I**) Analyzing properties of successfully transmitted uEPSCs, a significant decrease in uEPSC amplitude was evident in lesioned cultures when compared to non-lesioned controls (n°1_control_ = 144 events, n°1_3dp ECL_ = 82 events; n°2_control_ = 300 events, n°2_3dp ECL_ = 263 events; n°3_control_ = 426 events (one data point outside the axis limits), n°3_3dp ECL_ = 402 events; n°4_control_ = 486 events, n°4_3dp ECL_ = 364 events; n°5_control_ = 472 events, n°5_3dp ECL_ = 317 events; Mann-Whitney test for the respective pulse numbers). Individual data points are indicated by colored dots. Values represent mean ± s.e.m (* *p* < 0.05, *** *p* < 0.001, ns—nonsignificant difference).

**Figure 7 cells-13-00114-f007:**
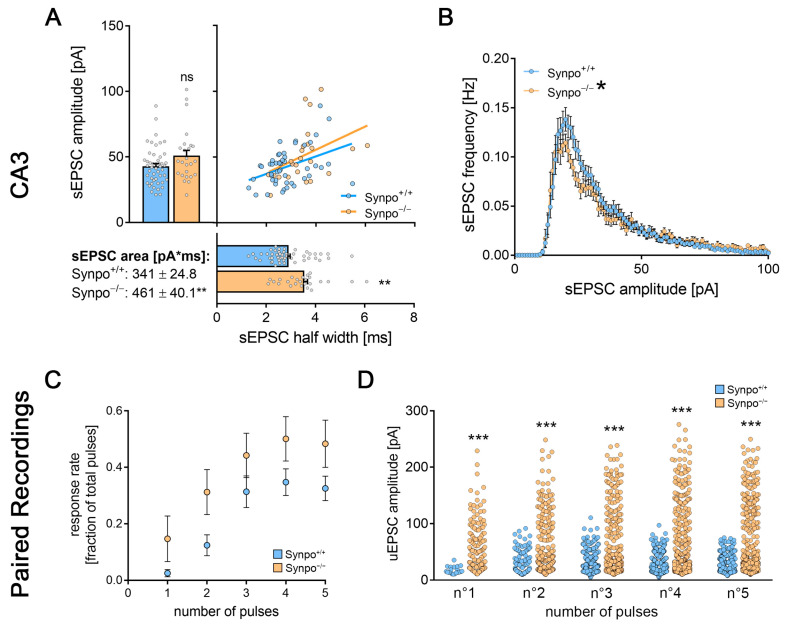
Synaptopodin deficiency shifted the hippocampal mossy fiber synapse to an excitable state under baseline conditions. (**A**,**B**) sEPSCs from CA3-PCs of both synaptopodin-deficient and wildtype tissue preparations, respectively. While the mean sEPSC amplitude was unchanged, sEPSC area and half width were significantly increased in synaptopodin-deficient tissue cultures ((**A**); n_Synpo+/+_ = 48; n_Synpo−/−_ = 25; Mann–Whitney test; Linear regression fit for XY-plot analysis; c.f., Figure 4 and Figure 6). Additionally, the sEPSC amplitude/frequency distribution was significantly changed between the two groups ((**B**); RM-two-way ANOVA followed by Sidak’s multiple comparisons test). (**C**,**D**) When comparing the response rate of postsynaptic CA3-PCs upon presynaptic action potential generation from paired recordings in Synpo^+/+^ and Synpo^−/−^ tissue preparations, no significant increase was observed ((**C**); n_Synpo+/+_ = 10; n_Synpo−/−_ = 12; Mann–Whitney test for the respective pulse numbers; c.f., Figure 3 and Figure 6). However, when analyzing the amplitudes of successfully transmitted uEPSCs at the postsynaptic CA3-PCs, a significant increase in uEPSC amplitudes became evident in synaptopodin-deficient tissue preparations after each of the five pulses respectively ((**D**); n°1_Synpo+/+_ = 20 events, n°1_Synpo−/−_ = 144 events; n°2_Synpo+/+_ = 115 events, n°2_Synpo−/−_ = 300 events; n°3_Synpo+/+_ = 292 events, n°3_Synpo−/−_ = 426 events (one data point outside the axis limits); n°4_Synpo+/+_ = 326 events, n°4_Synpo−/−_ = 486 events; n°5_Synpo+/+_ = 307 events, n°5_Synpo−/−_ = 472 events; Mann–Whitney test for the respective pulse numbers; c.f., Figure 4 and Figure 6). Individual data points are indicated by colored dots. Values represent mean ± s.e.m (* *p* < 0.05, ** *p* < 0.01, *** *p* < 0.001, ns—non-significant difference).

**Figure 8 cells-13-00114-f008:**
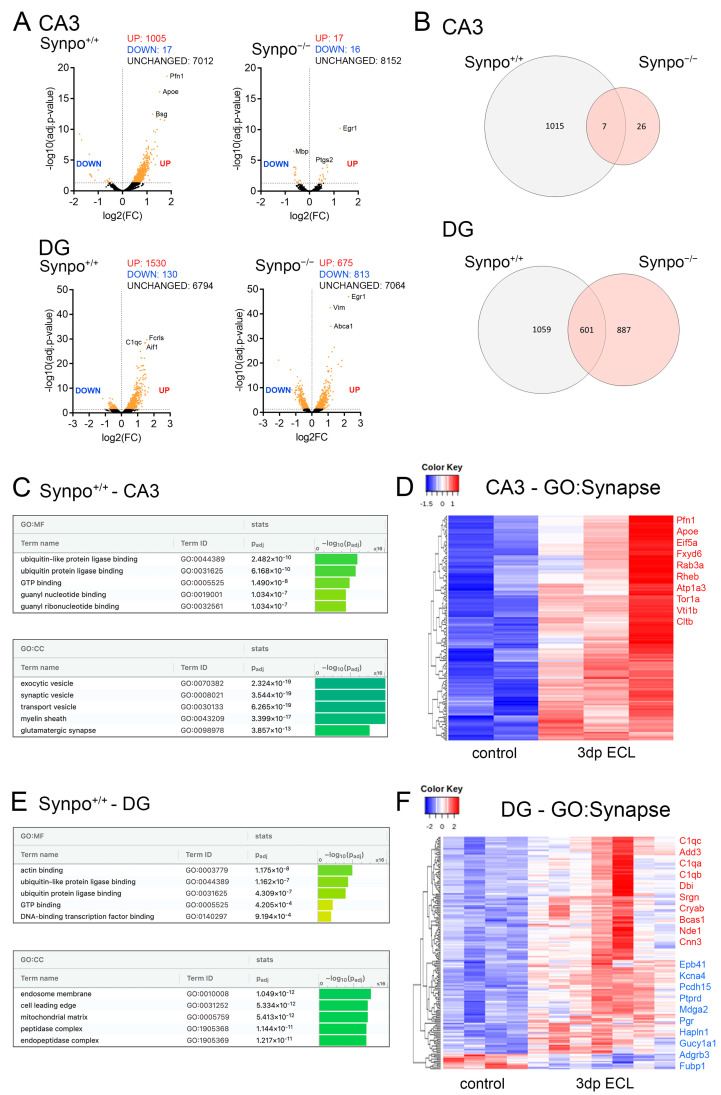
Entorhinal cortex lesion-induced synaptopodin-dependent transcriptomic changes. (**A**) Volcano plots visualize differential gene expression in wildtype (Synpo^+/+^) and synaptopodin-deficient (Synpo^−/−^) tissue preparations based on region-specific transcriptome analysis of the dentate gyrus (DG) and the CA3 area. Colored dots indicate differentially expressed genes (p_adj._ < 0.05; CA3-Synpo^+/+^: n_control_ = 2, n_3dp ECL_ = 3; DG-Synpo^+/+^: n_control_ = 4, n_3dp ECL_ = 7; CA3-Synpo^−/−^: n_control_ = 6, n_3dp ECL_ = 6; DG-Synpo^−/−^: n_control_ = 7, n_3dp ECL_ = 4; detailed DESeq2-outputs provided in Appendix A). (**B**) Venn-diagrams illustrate the magnitude of lesion-induced differential gene expression and the overlap in commonly regulated genes between wildtype and synaptopodin-deficient tissue preparations. Synaptopodin-deficient preparations showed deficits in lesion-induced gene expression mainly in the CA3 region. Despite the identification of commonly regulated genes, a considerable number of genotype-specific changes can be identified in DG samples. (**C**) For wildtype CA3 samples (Synpo^+/+^-CA3), a gene set enrichment analysis (gene ontologies: molecular function (MF) and cellular compartment (CC)) was performed and visualized (TOP-5) using the g:Profiler web platform. (**D**) Heatmap visualization demonstrates the differential expression of synapse related genes (GOCC: 0045202) upon lesion-induced cellular remodeling in the wildtype CA3 region. TOP-10 differentially expressed genes were additionally illustrated. (**E**) For wildtype DG samples (Synpo^+/+^-DG), a gene set enrichment analysis (gene ontologies: molecular function (MF) and cellular compartment (CC)) was performed and visualized (TOP-5) using the g:Profiler web platform. (**F**) Heatmap visualization demonstrates the differential expression of synapse related genes (GOCC: 0045202) upon lesion-induced cellular remodeling in the wildtype DG region. TOP-10 differentially expressed genes for both up- and downregulation were additionally illustrated. Analysis was performed on the Galaxy platform, and differential gene expression was determined using the DESeq2 application.

**Table 1 cells-13-00114-t001:** Entorhinal cortex lesion induced ultrastructural changes in postsynaptic densities in DG and CA3 subfields. Analysis of PSD areas in proximal and distal subfields of both DG and CA3 subfields. Kruskal–Wallis test followed by Dunn’s post hoc correction. Values represent median [interquartile range, 25th–75th percentile] (* *p* < 0.05, *** *p* < 0.001, ns—non-significant difference).

	Control	2hp ECL	3dp ECL
**DG proximal**	0.1 [0.07–0.2] μm^2^ (n_control_ = 183)	0.1 [0.08–0.1] μm^2^ (n_2hpl_ = 297) ns	0.09 [0.06–0.1] μm^2^ (n_3dpl_ = 231) ns
**DG distal**	0.2 [0.0975–0.3] μm^2^ (n_control_ = 62)	0.1 [0.09–0.2] μm^2^ (n_2hpl_ = 89) *	0.09 [0.06–0.1] μm^2^ (n_3dpl_ = 48) ***
**CA3 proximal**	0.1 [0.07–0.2] μm^2^ (n_control_ = 212)	0.1 [0.0685–0.2] μm^2^ (n_2hpl_ = 229) ns	0.1 [0.06–0.3] μm^2^ (n_3dpl_ = 224) *
**CA3 distal**	0.1 [0.08–0.2] μm^2^ (n_control_ = 100)	0.09 [0.07–0.1] μm^2^ (n_2hpl_ = 121) ns	0.08 [0.06–0.1] μm^2^ (n_3dpl_ = 120) ***

## Data Availability

Source files and detailed statistical analyses are available from the Dryad data repository (https://doi.org/10.5061/dryad.x0k6djhpc). Sequencing data have been deposited in the Gene Expression Omnibus (GEO) repository (accession number: GSE216509). Original data are available from the authors upon reasonable request.

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
