# Peer review of "Synaptopodin Regulates Denervation-Induced Plasticity at Hippocampal Mossy Fiber Synapses"

_cells, 2024, doi:10.3390/cells13020114_

Round 1
Reviewer 1 Report
Comments and Suggestions for Authors
Kruse and co-workers found lesion-induced changes in synaptic plasticity using mouse hippocampal slices by electrophysiological techniques. Furthermore, they examined synaptic fine structure and gene expression by transcriptome analysis to support the physiological changes. Especially, they also revealed an important role of Synaptopodin in synaptic plasticity. The authors used multiple advanced methods and report meaningful results in the manuscript. This reviewer believes the manuscript is valuable for other researchers in the field of neuroscience. However, information about the background lacks in the current one, and additional analysis and discussion are necessary for acceptance. My comments are below,
Comment 1.
Introduction and background: The authors need more explanation of the background of Synaptopodin in the introduction. This reviewer wonders why you suddenly refer Synaptopodin in line 63. Further information, especially a relationship of Synaptopodin with neurological diseases, is important, because you mention the diseases at the beginning in the introduction.
Comment 2.
Results and figures: In addition to synaptic volumes and ribosomes, the authors need to describe and analyze internal structure of pre- and post-synapses in the figure 3 and 5, such as synaptic vesicles and postsynaptic density. Additionally, it is hard to find the arrows in the TEM images in the figure 5. I recommend you to use colored arrows.
Comment 3.
Discussion: This reviewer has questions about phenotypes of Synaptopodin-deficient tissues. The phenotypes shown in the figure 7 (and probably figure 6) are at least partly caused by impairment of nervous development, such as neurogenesis, neuronal growth, and neural network formation, during embryonic, fetal, and neonatal stages. The authors need to discuss the situation of hippocampal neurons and circuit in deficient mice and to discuss its impact on your results. This comment is related with my comment 1.
Comment 4.
Discussion: The authors have to talk about a limitation of your methodology. (1) You used cultured tissues originating from neonatal, developing mice; thus, neurons and neural circuit in these tissues are immature. Nonetheless, you highlight a relationship of your results with Alzheimer’s disease and aging related cognitive decline in discussion, line 786–787. The process of these illnesses is stepwise degeneration of normal, mature neurons and neural circuit. This reviewer is wondering about how much accurate your experimental situation as a model of these illnesses could be. (2) Moreover, this reviewer is thinking that your experimental situation, especially in synaptopodin-deficient tissue, might be more similar to developmental disorder rather than aging related illnesses. Information about a relationship between the results you obtained and developmental disorder is important.(3) Regarding to your hypothesis at the last, how do you think about inputs coming from other brain regions, such as dopamine and noradrenaline in the midbrain and locus coeruleus respectively, for synaptic resilience? Because these inputs are completely excluded in your experiments, neurons and synapses might be more vulnerable to lesions. You might overestimate a role of Synaptopodin in synaptic plasticity.
Comment 5.
Conclusion: More importantly, the last is generally the conclusion on the basis of authors’ findings, not a new hypothesis containing references. You need to revise it to properly emphasize your study’s significance.
Author Response
Comment 1.
Introduction and background: The authors need more explanation of the background of Synaptopodin in the introduction. This reviewer wonders why you suddenly refer Synaptopodin in line 63. Further information, especially a relationship of Synaptopodin with neurological diseases, is important, because you mention the diseases at the beginning in the introduction.
Response: We thank the reviewer for this valuable comment. We have added an additional paragraph to the introduction to emphasize the role of synaptopodin and the spine apparatus organelle in regulating synaptic plasticity and its potential involvement in neuropsychiatric diseases.
Lines 48-55 now read: “At this synapse, previous studies have identified distinct morphological features, such as the regular occurrence of the spine apparatus organelle, which can be found in a subset of dendritic spines in telencephalic neurons [5]. Notably, the spine apparatus organelle and its associated actin-binding molecule synaptopodin were identified as crucial regulators of synaptic plasticity [6-8]. Moreover, recent work suggested a potential role for synaptopodin in neuropsychiatric diseases, such as cognitive decline and autism spectrum disorders [9, 10].”
Comment 2.
Results and figures: In addition to synaptic volumes and ribosomes, the authors need to describe and analyze internal structure of pre- and post-synapses in the figure 3 and 5, such as synaptic vesicles and postsynaptic density. Additionally, it is hard to find the arrows in the TEM images in the figure 5. I recommend you to use colored arrows.
Response: We thank the reviewer for this comment. In this study, we have focused on the role of the spine apparatus organelle – a postsynaptic internal structure – in regulating lesion-induced plasticity at hippocampal mossy fiber synapses. In Figure 5, therefore, we provide a detailed ultrastructural analysis of this organelle. For Figure 3, we have now added further analyses of postsynaptic densities (PSDs, see Table 1), which corresponds to our findings on lesion-induced changes in synaptic volume.
For the dentate gyrus, lines 449-453 now read:
“In addition, ECL caused alterations in the surface area of postsynaptic densities in the DG (PSDs, Table 1): While the surface area of PSDs remained unchanged at proximal dendrites, it decreased significantly at distal dendrites after ECL.”
For CA3, lines 499-501 now read: “Moreover, the surface area of PSDs increased significantly at proximal synapses in the CA3 subfield 3 days after lesion. At the same time, at the distal synapses there was a significant decrease in PSD surface area (Table 1).”
Moreover, we increased the visibility of the arrows according to your suggestions. Thank you again!
Comment 3.
Discussion: This reviewer has questions about phenotypes of Synaptopodin-deficient tissues. The phenotypes shown in the figure 7 (and probably figure 6) are at least partly caused by impairment of nervous development, such as neurogenesis, neuronal growth, and neural network formation, during embryonic, fetal, and neonatal stages. The authors need to discuss the situation of hippocampal neurons and circuit in deficient mice and to discuss its impact on your results. This comment is related with my comment 1.
Response: Thank you for this comment! Previous work has demonstrated that synaptopodin protein expression starts postnatally (Czarnecki et al., 2005). We are therefore convinced that our findings rely on plasticity defects in synaptopodin-deficient tissue cultures rather than developmental alterations in neuronal networks. We now emphasize this point in the revised version of this manuscript. Moreover, we have included a further note of the phenotype in synaptopodin-deficient animals.
Lines 843-848 now read: “The spine apparatus organelle, which is lacking in synaptopodin-deficient mice [20], was previously linked to learning deficits in adult mice and alterations in synaptic plasticity in various research models [75]. Of note, synaptopodin expression and spine apparatus formation start postnatally [76]. Therefore, synaptopodin-mediated plasticity regulation in mature systems seems to have a substantial contribution to the observed phenotypes.”
Comment 4.
Discussion: The authors have to talk about a limitation of your methodology. (1) You used cultured tissues originating from neonatal, developing mice; thus, neurons and neural circuit in these tissues are immature. Nonetheless, you highlight a relationship of your results with Alzheimer’s disease and aging related cognitive decline in discussion, line 786–787. The process of these illnesses is stepwise degeneration of normal, mature neurons and neural circuit. This reviewer is wondering about how much accurate your experimental situation as a model of these illnesses could be. (2) Moreover, this reviewer is thinking that your experimental situation, especially in synaptopodin-deficient tissue, might be more similar to developmental disorder rather than aging related illnesses. Information about a relationship between the results you obtained and developmental disorder is important.(3) Regarding to your hypothesis at the last, how do you think about inputs coming from other brain regions, such as dopamine and noradrenaline in the midbrain and locus coeruleus respectively, for synaptic resilience? Because these inputs are completely excluded in your experiments, neurons and synapses might be more vulnerable to lesions. You might overestimate a role of Synaptopodin in synaptic plasticity.
Response: We thank the reviewer for the thoughtful comments to our discussion.
- Organotypic entorhino-hippocampal tissue cultures are characterized by in vivo-like connectivity (Frotscher and Heimrich, 1993; Hildebrandt-Einfeldt et al., 2021; Lenz et al., 2023), mature cell states (Leutgeb et al., 2003; Radic et al., 2017) and steady-state inflammatory conditions (Delbridge et al., 2020). Recent work has demonstrated the similarities between mossy fiber synapses in organotypic tissue cultures and in vivo (Maus et al., 2020). Moreover, the translational relevance for adult murine and human neural circuits has been demonstrated. However, we addressed the limitation of using tissue cultures in this study in the revised manuscript. Lines 780-782 now read: “However, projections from other cortical and subcortical regions and other environmental factors are missing in in vitro systems that might influence synaptic transmission and plasticity in the hippocampus [52, 53].”
- We addressed this comment in the revised version of our discussion Lines 843-848 now read: “The spine apparatus organelle, which is lacking in synaptopodin-deficient mice [20], was previously linked to learning deficits in adult mice and alterations in synaptic plasticity in various research models [75]. Of note, synaptopodin protein expression and spine apparatus formation start postnatally [76]. Therefore, synaptopodin-mediated plasticity regulation in mature systems seems to have a substantial contribution to the observed phenotypes.”
- Thank you for raising this interesting point. Indeed, the impact of inputs from other brain regions cannot be elaborated in these tissue culture preparations. We therefore agree with the reviewer that this limitation should be mentioned in the discussion of our manuscript. Lines 780-782 now read: “However, projections from other cortical and subcortical regions and other environmental factors are missing in in vitro systems that might influence synaptic transmission and plasticity in the hippocampus [52, 53].”
Comment 5.
Conclusion: More importantly, the last is generally the conclusion on the basis of authors’ findings, not a new hypothesis containing references. You need to revise it to properly emphasize your study’s significance.
Response: We have now added a conclusion at the end of this manuscript.
Lines 862-867 now read: “In conclusion, our results demonstrate a role for synaptopodin in regulating synaptic transmission and lesion-induced plasticity at the hippocampal mossy fiber synapse in organotypic entorhino-hippocampal tissue cultures. Our data indicate a relationship between ribosomes and the spine apparatus organelle, which suggests that synaptopodin and the spine apparatus organelle might be involved in synapse related local protein synthesis.”
Reviewer 2 Report
Comments and Suggestions for Authors
Kruse P ., et al elucidate the impact of entorhinal cortex lesions in mouse brain tissue cultures, revealing that partial denervation strengthens excitatory neurotransmission in dentate granule cells and CA3 pyramidal cells. The authors identify synaptopodin and the spine apparatus organelle as crucial elements in regulating lesion-induced synaptic plasticity at hippocampal mossy fiber synapses.
The authors utilized entorhinal cortex lesions in organotypic entorhinal-hippocampal tissue cultures to investigate synaptic changes in the dentate gyrus and CA3 network following partial denervation. Furthermore, the findings reveal ultrastructural alterations and enhanced excitatory synaptic transmission in both dentate granule cells (dGCs) and CA3 pyramidal cells (CA3-PCs), suggesting synaptopodin and the spine apparatus organelle as crucial players in regulating lesion-induced synaptic plasticity, with a potential association with dendritic ribosomes near synaptic sites. According to the authors, Synaptopodin-deficient cultures lacking the spine apparatus organelle displayed impaired expression of lesion-induced plasticity, highlighting the novelty of this investigation - the organelle's essential role.
A previous study from the author's group has already shown the crucial role of synaptopodin in synaptic plasticity (https://www.pnas.org/doi/10.1073/pnas.1213677110). Here the authors showed increased excitatory synaptic strength in both dentate granule cells (dGCs) and CA3 pyramidal cells (CA3-PCs) post-lesion, accompanied by alterations in the region-specific transcriptome, identifies denervation-induced adaptations at hippocampal mossy fiber synapses and highlights the regulatory role of synaptopodin and the spine apparatus organelle in orchestrating lesion-induced plasticity.
According to the authors, the interaction between ribosome-associated synaptopodin & the spine apparatus organelle either direct or indirect, remains uncertain, and co-immunoprecipitation could potentially be targeting mRNA transport/storage granules. The overall speculation points towards local protein synthesis near synaptopodin clusters.
The experiments designed for this study are justified and the results are significant. The introduction and the discussion were written clearly with proper information and references. High appreciation should go to the authors as they mention animal ethics in the methods.
Nonetheless, the article seemed to possess good value toward lesion-induced synaptic plasticity, where the entorhinal cortex lesion elicited alterations in the regional transcriptome, potentially playing a role in the instantiation of lesion-induced synaptic plasticity.
Overall, the clarity of the text needs very few readjustments. The manuscript has minor typographical and grammatical errors. The results and the figures were consistent based on the written legends and results. The quantitative analyses are much appreciated but need more careful revision based on the significance of the data. In general, the manuscript can accomplish the caliber of quality for consideration for publication in the Journal of “Cells” with minor changes. The authors are advised to consider the comments below:
Comments:
1. Figure 1D- H, Figure 7D/ Please mention the names of the comparable groups in each graph.
2. Could you please provide direct evidence to support the statement “Synaptopodin and the spine apparatus organelle form associations with dendritic ribosomes.” Like: co-immunostaining (IHC)
3. Could you please give a brief outline of how you analyze synaptic complexity, which is the number of postsynaptic sites per presynaptic bouton, at proximal synapses in DG?
4. It would be helpful to understand if you could show with co-IP /western blotting the interaction between Synaptopodin and ribosomal complexes. Or by using any synaptic proteome analysis.
5. How do you explain the function of Synaptopodin in local protein synthesis which may involve synaptic plasticity?
Comments on the Quality of English LanguageOverall, the clarity of the text needs very few readjustments. The manuscript has minor typographical and grammatical errors.
Author Response
- Figure 1D- H, Figure 7D/ Please mention the names of the comparable groups in each graph.
Response: Thank you for this point. We have changed the figures according to your suggestions and did so for Figure 6H as well.
- Could you please provide direct evidence to support the statement “Synaptopodin and the spine apparatus organelle form associations with dendritic ribosomes.” Like: co-immunostaining (IHC)
Response: In this study, we provide several lines of evidence for direct or indirect associations of synaptopodin / the spine apparatus organelle and ribosomes (Figure 5):
- Transmission electron microscopy analysis of synaptic sites reveal the preferential presence of ribosomes – which can be detected due to their characteristic morphology – in dendritic spines that contain a spine apparatus organelle. Therefore, the presence of the spine apparatus organelle is associated with the localization of ribosomes near synaptic sites. We clarify this point in the revised version of our manuscript:
Lines 75-78 now read: “We therefore conclude that the spine apparatus organelle is essential for regulating lesion-induced plasticity, which might be linked to the localization of ribosomes near synaptic sites.”
- Negative-contrast electron microscopy of synaptopodin coIP samples suggest an association with filamentous structures and ribosomes. However, it remains unclear whether direct or indirect interactions link synaptopodin and ribosomes.
- Synaptopodin-coIP samples contain numerous synaptic mRNAs, which suggests a functional relevance of synaptopodin ribosome interactions for synaptic transmission.
Lines 550-552 now read: “Therefore, these data hint towards a direct or indirect functional association between synaptopodin/the spine apparatus organelle and ribosomes, which might contribute to lesion-induced synaptic plasticity through local protein synthesis.”
- Could you please give a brief outline of how you analyze synaptic complexity, which is the number of postsynaptic sites per presynaptic bouton, at proximal synapses in DG?
Response: We have added further descriptions in our materials and methods section. Lines 285-289 now read:
“In order to analyze synaptic complexity, the images with the 3D-reconstructed synaptic compartments were used. The number of postsynaptic compartments and their postsynaptic densities were manually counted for each respective presynaptic compartment by an investigator blinded to experimental conditions.”
- It would be helpful to understand if you could show with co-IP /western blotting the interaction between Synaptopodin and ribosomal complexes. Or by using any synaptic proteome analysis.
Response: A proteome analysis for synaptopodin interactions has been previously performed (Falahati et al., 2022). In this analysis, ribosomal proteins were indeed identified as interacting partners. This is in line with our results that find an interaction with mRNAs. We therefore conclude a direct or indirect interaction between synaptopodin and ribosomes. This point is mentioned in the discussion of our manuscript.
Lines 769-772 read: “Together with findings that indicate interactions between synaptopodin, ribosomal subunits and ER-related proteins [45], we suggest a role for synaptopodin as local regulator of synaptic plasticity at hippocampal mossy fiber synapses.”
- How do you explain the function of Synaptopodin in local protein synthesis which may involve synaptic plasticity?
Response: Thank you for emphasizing this point. Throughout the revised version of our manuscript, we point out that synaptopodin might support local protein synthesis by assembling ribosomes near synaptic sites. Specifically, Figure 3 demonstrates that the presence of the spine apparatus organelle in dendritic spines is associated with the localization of ribosomes near synaptic sites.
Reviewer 3 Report
Comments and Suggestions for Authors
The authors have chosen a very clear subject and performed elegant organ-typical studies and subsequent biochemical assessments including RNAseq and electron microscopy that provides clear indication for the role of synaptopodin the the denervation associated functional changes in synaptic neurotransmission.
The scientific premise of the study explores the unmet need of how denervation that follows demyelination, traumatic injury or cell death in various neuropathological states occurring either developmentally or in pathological aging impinges on the spine apparatus in the vulnerable region of the entorhinal cortex. By studying organotypic cultures where the authors remove this region, they looked at the mossy fiber synapses that is crucial for spatial learning. They assessed the cellular learning disabilities using paired recordings from the dentate granule cells to the CA3 regions and looked at excitatory neurotransmission. When the explored the biochemistry underlying the effect on dendritic spines, they were able to appreciate a role for synaptopodin. There is a gap in the understanding of how denervation affects synaptic neurotransmission - this manuscript has done a thorough job of studying a specific synaptic connection and the effect of denervation that would be induced via injury or insult that they modeled quite elegantly and demonstrated the effect on an underlying biochemical pathway. The subject matter is unique because it utilizes some distinct approaches that are designed to explore the mechanism very effectively. The authors have managed a very exhaustive study that is quite complete for the subject matter and the scientific rationale that they explored. Nevertheless, it is important to emphasize that the mechanism of action and the effect observed may be specific to the entorhinal ablation on the mossy fiber pathway unless there are future studies to corroborate that the mechanism of action is conserved for various excitatory synapses. Using a solid scientific premise addressing a specific question of concern, the authors have managed to provide a very cogent and complete study that can be published as is since it advances the field a little further. As it stands, the material is complete and exhaustive for both the manuscript length and interest.
Author Response
We thank this reviewer for acknowledging the quality and the scope of our work.Round 2
Reviewer 1 Report
Comments and Suggestions for Authors
The authors have clearly answered my questions and revised their manuscript in accordance with my comments. This reviewer believes the revised one be published.